# Estimating the effect of social inequalities on the mitigation of COVID-19 across communities in Santiago de Chile

Nicolò Gozzi[1], Michele Tizzoni [2], Matteo Chinazzi[3], Leo Ferres [4,5], Alessandro Vespignani[2,3] & Nicola Perra[1,3 ✉]

We study the spatio-temporal spread of SARS-CoV-2 in Santiago de Chile using anonymized mobile phone data from 1.4 million users, 22% of the whole population in the area, characterizing the effects of non-pharmaceutical interventions (NPIs) on the epidemic dynamics. We integrate these data into a mechanistic epidemic model calibrated on surveillance data. As of August 1, 2020, we estimate a detection rate of 102 cases per 1000 infections (90% CI: [95–112 per 1000]). We show that the introduction of a full lockdown on May 15, 2020, while causing a modest additional decrease in mobility and contacts with respect to previous NPIs, was decisive in bringing the epidemic under control, highlighting the importance of a timely governmental response to COVID-19 outbreaks. We find that the impact of NPIs on individuals' mobility correlates with the Human Development Index of comunas in the city. Indeed, more developed and wealthier areas became more isolated after government interventions and experienced a significantly lower burden of the pandemic. The heterogeneity of COVID-19 impact raises important issues in the implementation of NPIs and highlights the challenges that communities affected by systemic health and social inequalities face adapting their behaviors during an epidemic.

[1] Networks and Urban Systems Centre, University of Greenwich, London, UK. [2] ISI Foundation, Turin, Italy. [3] Laboratory for the Modeling of Biological and Socio-technical Systems, Northeastern University, Boston, MA, USA. [4] Data Science Institute, Universidad del Desarrollo, Santiago, Chile. [5] Telefónica R&D, Santiago, Chile. ✉email: n.perra@greenwich.ac.uk

As of September 1, 2020, Chile has reported more than 400,000 cases and 590 SARS-CoV-2 deaths per million, becoming one of the worst COVID-19 epidemic globally[1]. Officially, the first SARS-CoV-2 case in Chile was detected on March 3, 2020[2]. Although other cases were rapidly confirmed all over the country, the urban area of the capital city, Santiago Metropolitan Region, quickly became the epicenter of the national epidemic. Indeed, as of September 1, 2020, about 70% of the total cases in the nation have been reported in the comunas (i.e., municipalities) of Santiago, making it one of the largest urban COVID-19 outbreak in the world. The first set of non-pharmaceutical interventions (NPIs) were put in place in mid-March, when schools were closed, public gatherings were banned, and passengers traveling from high-risk countries were mandated to self-isolate for 14 days. However, the adopted measures were not able to contain the contagion: after a sharp increase in cases, a full lockdown was instituted to the whole Metropolitan Region on May 15[3].

In this work, we model the spatial and temporal spread of SARS-CoV-2 in 37 comunas of the urban area of Santiago which comprises 6.4 million individuals. We aim to provide a data-driven characterization of the unfolding of the COVID-19 epidemic and an estimation of the impact of NPIs on its spreading. Research on similar geographical scales has been conducted, for example, for Boston[4], Wanzhou[5], New York City[6], and London[7]. While COVID-19 is clearly a global issue, the measures adopted to mitigate and/or suppress the spread of SARS-CoV-2 have been quite heterogeneous across countries and often across sub-regions within the same country[8–10]. Hence, modeling specific contexts and spatial scales is key to identify separate effects of NPIs, their practical impact when revoked, and possibly reintroduced over an extended period of time. We study the reduction of mobility and contacts inferred from mobile devices as input for a spatially and age-structured epidemic model. In fact, mobile device data can be used to evaluate, in near real-time, the effects of interventions and self-initiated behavioral changes on the mobility of people and to inform large scale epidemic models[11–15]. Here, we use anonymised data provided by a major mobile phone operator in South America (Telefónica Movistar), with a market share of 24.61% as of March 2020.

To characterize the changes in mobility and physical contacts during the outbreak, we used anonymised data from 1.4 million mobile devices (about 22% of the total population in the comunas under consideration). We find consistent downward trends coinciding with the NPIs issued by local and national authorities. We estimate that the first set of NPIs issued on March 16 led to a reduction of about 48% in the number of travels between comunas. An additional 17% reduction is observed with the introduction of the full lockdown on May 15.

We develop a stochastic mechanistic epidemic model integrating mobility, physical contacts, and census data. The model suggests that the full lockdown, while causing a modest additional decrease in mobility and physical contacts with respect to the NPIs already in place, was decisive in bringing the epidemic under control. This relatively small additional decrease in mobility and contacts was enough to push the effective reproductive number below the critical value of 1, a clear example of the threshold effects characterizing epidemic dynamics on structured mobility networks[16]. We estimate that the full lockdown prevented an additional 34.7% (95% CI: [27.2%, 44.1%]) increase in the total number of deaths. In addition, we estimate the critical impact of the timing of the full lockdown through counterfactual scenarios: an additional week of delay would have corresponded to an 18.1% (95% CI: [6.0%, 34.0%]) more intense incidence peak according to our estimates.

Despite being regarded as a high-income country, Chile and its capital city show concerning social and economic inequalities[17]. Unfortunately, this makes Santiago a natural experiment to investigate the link between socioeconomic disparities and the burden of the pandemic. We explore this important dimension finding that changes in mobility patterns strongly correlate with the economic and development indicators of comunas. Furthermore, the model links these observations with heterogeneous burden of COVID-19 across comunas which is also observed in the epidemiological data reported by the national surveillance. More precisely, our results suggest higher attack and death rates in disadvantaged areas. Due to challenges faced in reducing their mobility and contacts, communities exhibiting systemic social disparities are affected in a differential way by government-mandated NPIs and disease's burden. This observation raises the key issue of health disparities in the management of emerging infectious diseases such as COVID-19.

## Results

We evaluate the effects of NPIs policies, government-mandated mobility limitations by integrating mobile phone data and an epidemic model. We identify three phases of the epidemic management in Santiago: (i) before March 16 (business as usual, baseline), (ii) between March 16 and May 15 (first set of NPIs), and (iii) after May 15 (full lockdown). For convenience, we will refer to the period March 16 to May 15 as the partial lockdown and to the period after May 15 as the full lockdown.

It is important to notice how the timeline of interventions is fairly complex. It includes night curfews, dynamic quarantine, and lockdowns restricted to a few comunas across the region studied here and in other parts of Chile[2]. However, as we see below, the data suggest that those measures did not have a significant impact on people's behaviors, thus for simplicity we consider the two main sets of NPIs only. We characterize the three phases outlined above in terms of (a) mobility among comunas, and (b) contacts reduction between individuals. Mobility describes the (varying) rates at which people travel among different comunas, while contacts reduction parameters estimate to what extent physical contacts varied in time in each comuna (more details in the "Methods" section).

**Effects of NPIs and social inequalities.** In Fig. 1a, we provide an overview of mobility in Santiago during the period of study. As a proxy for general mobility, we consider the number of devices visiting a comuna that is different from their home one (see "Methods" section). We observe a sharp drop following the first set of interventions on March 16. Afterward, mobility remains fairly constant until the introduction of the full lockdown on May 15, when we observe an additional 17% decrease. As we will show below, this intervention represented an important tipping point of the epidemic in Santiago.

More in detail, we represent changes in mobility flows across comunas in Fig. 1b. The partial lockdown causes an average drop of about 48%, while, with the introduction of the full lockdown, mobility across comunas drops by 65% with respect to the baseline. For each comuna we also consider the mean percentage decrease in mobility after March 16 and compare it with the Human Development Index (HDI), a coefficient that measures key aspects of human development, such as life expectancy, education, and per capita income[18] (see the "Methods" section for more details on HDI calculation). In Fig. 1c, we observe that a greater decrease in mobility is generally associated with a higher HDI (Pearson correlation coefficient $\rho = -0.80$, $p < 0.001$). The same trend is observed in

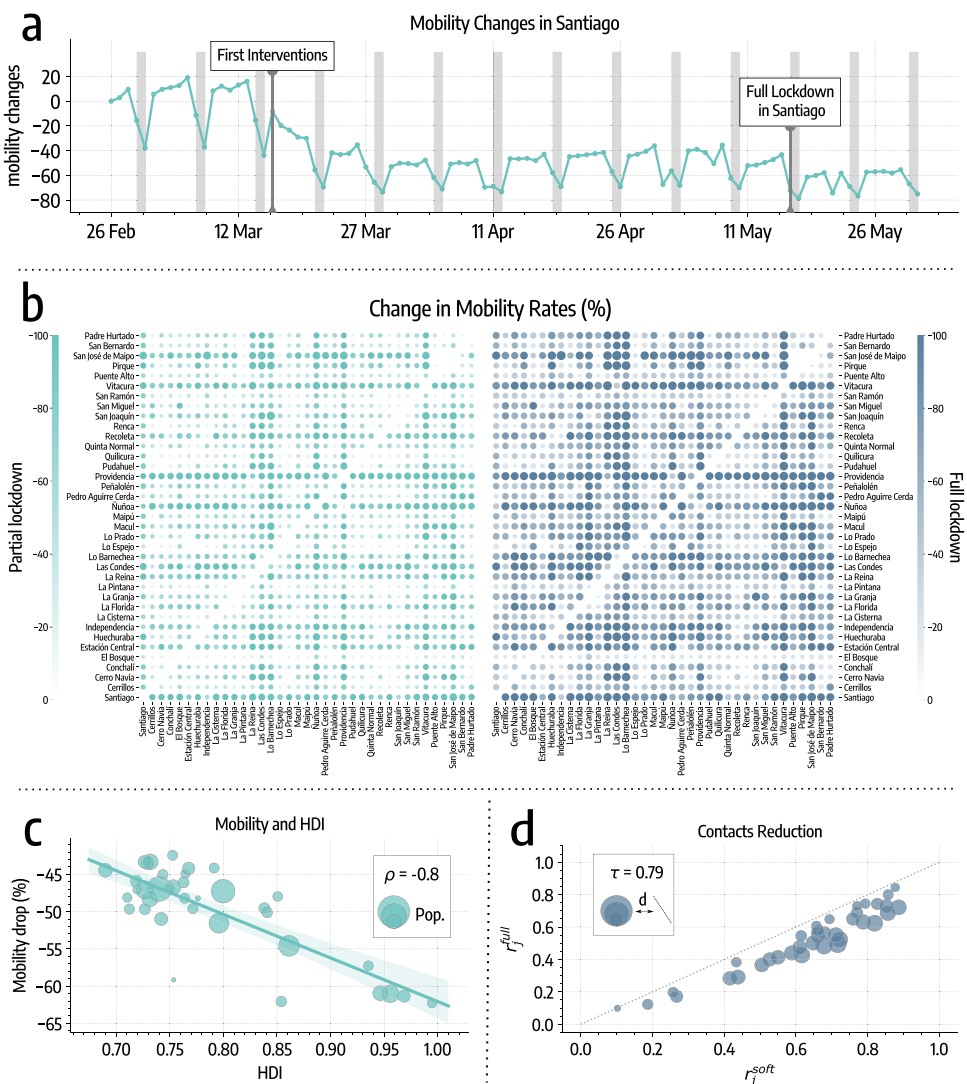

**Fig. 1 Mobility and contacts changes in Santiago. a** Overview of mobility changes, we consider the number of devices visiting a comuna different from their home one as a proxy for general mobility (gray areas represent weekends). Changes are expressed as percentages with respect to February 26. **b** Percentage changes in mobility rates (with respect to mobility before March 16). On the left drop-in mobility after the partial lockdown, on the right after the full lockdown. Color and dots size are scaled according to the magnitude of the change. **c** Average percentage mobility decreases after March 16 versus HDI of different comunas. We display the regression line, 95% CI, and the Pearson correlation coefficient $\rho$. Dots size is proportional to the population of the comuna. **d** Scatter plot of contacts reduction parameters during partial ($r_j^{partial}$) and full ($r_j^{full}$) lockdown. We display the Kendall rank correlation coefficient $\tau$. Dots size is proportional to the distance from the diagonal (bigger dots indicate comunas where contacts decreased more after the full lockdown).

the absolute change of mobility (see Supplementary Fig. 5) suggesting that wealthier and more developed comunas became more isolated after the interventions. This result is in line with previous studies that showed how changes in mobility patterns following government-issued interventions, and the extent to which people can afford social distancing, vary across different socio-demographic groups[15,19–22].

In Fig. 1d, we represent contacts reduction parameters. Across the board, contacts drop by 36% with the first set of NPIs policies and by an additional 11% with the lockdown. Since all points are below the diagonal, we conclude that, with the introduction of the full lockdown, contacts decrease further in all comunas. Also, the decrease is consistent with the existing reductions after the partial lockdown. Indeed $r_j^{partial}$ and $r_j^{full}$ show a high significant correlation (Kendall rank correlation coefficient $\tau = 0.79$, $p < 0.001$).

**The spread of COVID-19 in Santiago.** We use the mobility data to develop and inform a stochastic mechanistic metapopulation epidemic model (see "Methods" for details) and simulate the spread of COVID-19 in the comunas of Santiago. The model is calibrated on official surveillance data and takes as initial seeding the realistic projections of active cases on March 1, 2020, in the Metropolitan area of Santiago from ref. [14].

We use an Approximate Bayesian Computation (ABC) approach[23,24] (see details in the "Methods" section) to find the posterior distribution of the reproductive number in Santiago (median $R_0 = 2.66$, 95% CI: [2.58, 2.72]), which is in line with previous findings that identify the value of $R_0$ of SARS-CoV-2 to be in the range between 2 and 3 in different countries[25–27]. In Fig. 2a, we report the number of weekly deaths projected by the model together with official figures (used for calibration). The two time series show a good agreement with a high correlation

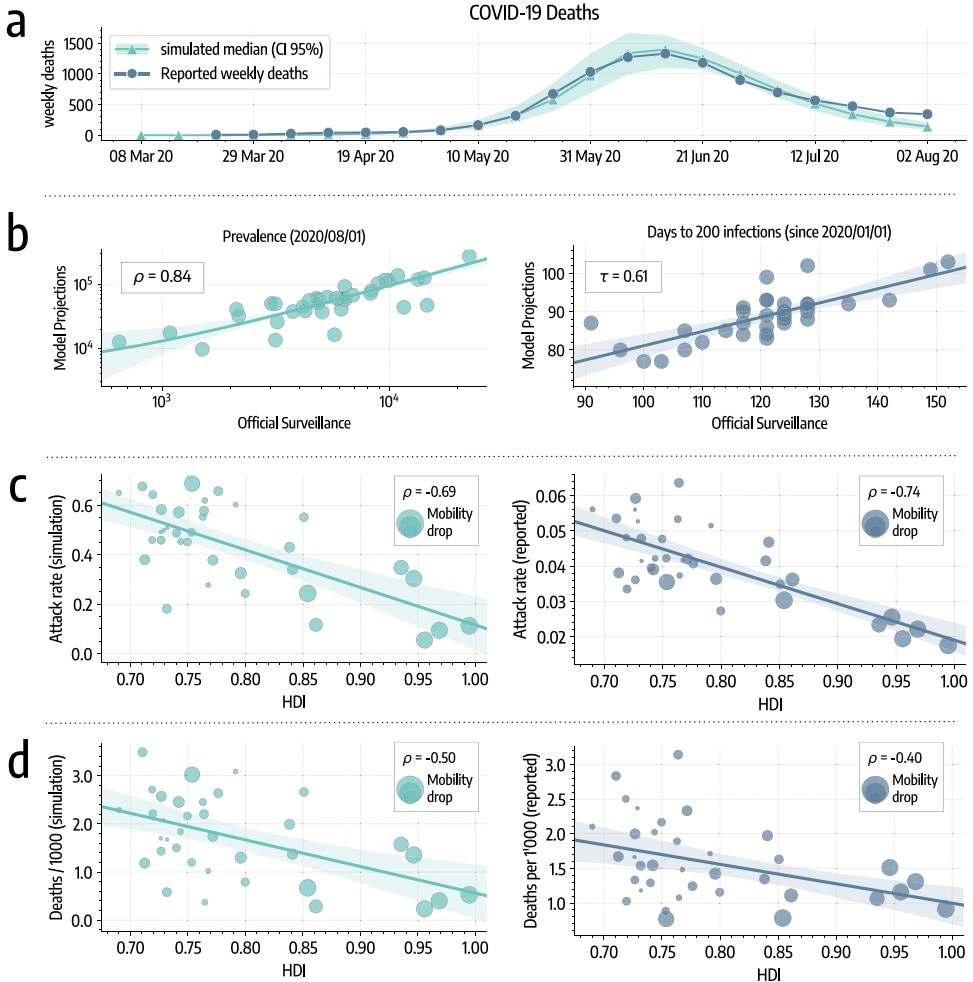

**Fig. 2 SARS-CoV-2 spreading in Santiago. a** We represent the simulated (median and 95% CI) and reported weekly deaths used for model calibration. **b** Left: scatter plot of reported versus simulated cases as of August 1, 2020. Right: scatter plot of days (since January 1, 2020) needed to reach 200 infections in each comuna as reported by official surveillance and as projected by our model. **c** Scatter plot of HDI versus attack rate as of August 1, 2020, in different comunas as projected by our model (left) and as reported by official surveillance (right). Size of dots is scaled according to the mobility drops after March 16 (bigger bullets indicate bigger decreases in mobility). **d** Scatter plot of HDI versus deaths per 1000 as of August 1, 2020, in different comunas as projected by our model (left) and as reported by official surveillance (right). Size of dots are scaled according to mobility drops after March 16. In panels **b**, **c**, and **d** we show regression lines, 95% CI, and Pearson correlation coefficient $\rho$ or Kendall rank correlation coefficient $\tau$.

($\rho = 0.99$, $p < 0.001$) and a median absolute percentage error of 12%. Interestingly, the agreement between data and model starts to deviate in the last two data points (end of July early August). We can speculate that, at least in part, this might be due to a lockdown fatigue. In fact, while the official restrictions were relaxed later (mid of August), a decrease in compliance, linked to the reduction of cases/deaths, could have taken place earlier. The period is outside our current data coverage. Hence, we leave testing such hypothesis to future work.

As of August 1, 2020, the median projected fraction of infected individuals in the area under study is 38.7% (95% CI: [35.1%, 41.6%]). This estimate is about tenfold the official reported figures. We are not aware of publicly available seroprevalence studies that we can use as a comparison and validation. For this reason, we can only consider qualitative evidence hinting that the Chilean outbreak has affected a significant fraction of the population. For example, the share of positive COVID-19 tests peaked at 59.10% on June 18, and in the Santiago Metropolitan Region the occupation of ICU beds reached almost saturation level (95%) in May. Also, a recent epidemiological study aimed at characterizing the first wave in Chile showed significant under-reporting of symptomatic cases (around 50%) based on estimates

of the Case Fatality Rate[2]. Another recent modeling study focused on the COVID-19 outbreak in Santiago estimates a number of infected individuals 5–10 times larger than the official figures[28]. Finally, previous seroprevalence studies conducted, for example, in the United States[29], Spain[30], Italy[31], Brazil[32], and Iran[33], showed that the actual number of COVID-19 infections is several times (factors vary from 4 to 20) those reported by the official surveillance. As a sensitivity check, we repeated the calibration considering the upper limit of the 95% credible interval for the Infection Fatality Rate from ref. [34]. This leads to a projected median prevalence of 28.3% (95% CI: [23.5%, 32.3%]) but to a sensibly worse fit of the data ($\rho = 0.82$, median percentage error of 49%).

Projected cases present a significant correlation with official numbers as can be observed in Fig. 2b ($\rho = 0.84$, $p < 0.001$). Besides, we compare the dates when 200 infections have been reached in different comunas according to our model and official surveillance, finding a significant correlation (Kendall rank correlation coefficient $\tau = 0.61$, $p < 0.001$). Similar results are found considering instead the dates when 50, 100, and 500 cases have been reached (see Supplementary Fig. 6). Interestingly, the same dates estimated through modeling are much earlier, hinting

that many of the infections in the initial phase of the spreading went unreported. In Fig. 2c, we show the attack rates versus the HDI of different comunas. We find a strong correlation between attack rates computed on officially reported cases and HDI ($\rho = -0.74$, $p < 0.001$), providing evidence that wealthier comunas experienced significantly smaller outbreaks. In addition, the very same picture emerges from our modeling results. Indeed, simulated attack rates present a high significant correlation with HDI ($\rho = -0.69$, $p < 0.001$). Finally, in Fig. 2d we show the number of deaths per 1000 versus the HDI of different comunas. We find a significant correlation between HDI and both simulated ($\rho = -0.50$, $p < 0.002$) and officially reported ($\rho = -0.40$, $p < 0.02$) deaths, hinting that wealthier comunas experienced also a smaller burden in terms of casualties. We note that the correlations obtained in this case are lower than those found previously for the attack rates. This may be due to the interplay between diverse age distributions and age-dependent mortality rates. Indeed comunas with higher HDI have a higher mean age and the infection fatality rate for COVID-19 is significantly higher in older age brackets.

In Section 2 of the Supplementary Information, we compare our modeling approach and results with three simpler models. The first neglects the mobility between comunas. In other words, we attempt to fit the evolution of deaths in the region by considering each comuna as a separate population. Interestingly, this approach leads to a median absolute percentage error of 38% which implies a worse performance than the main model discussed above (12%). This result highlights the importance of capturing the coupling between areas and accounting for spatio-temporal heterogeneities in spreading patterns which are shaped, among other factors[35], by mobility. The second and third consider the entire metropolitan area as a single, age-structured, population. Similar approaches have been proposed to model the spreading of SARS-CoV-2 and the impact of NPIs in cities[7], regions[15], and countries[36]. The difference between these two models lays on the data we used to capture the effects of NPIs on contact matrices. In one, we use estimates measured from the mobile phone data. In the other instead, we use the Google Mobility Reports[37] and the Oxford COVID-19 Government Response Tracker[38,39] (see Section 2 of the Supplementary Information for details). Both models lead to inferior performance, with a median absolute percentage error of 18% and 43%, respectively.

Overall, out of these three simpler models, the second is the closest to the performance of our main modeling approach described above. However, by construction, it does not provide any information about the heterogeneous spread and impact of the virus across comunas.

**Counterfactual scenarios**. To assess the impact of heterogeneous responses to the spreading we run a hypothetical scenario in which mobility and contacts decrease uniformly across comunas, starting on March 16. More in detail, we apply to all comunas the average reduction in mobility and contacts observed for the 4th quartile of HDI (i.e., 25% comunas with higher HDI). See Section 3 of the Supplementary Information for more details. According to our simulations, this leads to a significant decrease of cases and deaths: −83.8% (95% CI: [−77.6%, −88.6%]) fewer cases and −70.5% (95% CI: [−55.0%, −80.9%]) fewer deaths as of May 15, the date when the full lockdown was enforced. Interestingly, the uniform reduction we are imposing implies a relatively modest additional decrease in mobility and contacts with respect to the ones estimated through mobile phone data. In this hypothetical scenario, with the partial lockdown mobility rates drop by 55% and contacts by 49% (versus, respectively, the 48%

and the 36% estimated in our main analysis). Although such homogeneous reduction across comunas is a theoretical exercise that does not consider complex socioeconomic constraints, that go from the collective need to keep key supply chains active to the individual imperative to feed their own family, it crystallizes the dramatic effects of inequality on disease spreading on the one side, and it shows the positive benefits of equal, early, and strong responses on the other.

We also use the model to investigate counterfactual scenarios aimed at estimating the impact of NPIs on the spread of COVID-19 in Santiago. As a first counterfactual scenario, we simulated the epidemic in the absence of a full lockdown. From Fig. 3a, we observe that this leads on average to a 21.6% (95% CI: [7.5%, 41.3%]) more intense incidence peak and 34.7% (95% CI: [27.2%, 44.1%]) more deaths. To estimate the impact of the timing of the full lockdown, we run simulations where we anticipate or delay it up to 4 weeks. According to results in Fig. 3b, an earlier lockdown implies a less intense incidence peak (from around −20% to −35%). It is interesting to note, however, that a delay of 1–2 weeks has a very similar effect of no intervention at all in terms of incidence peak intensity. More specifically, 1 week delay causes a 18.1% (95% CI: [6.0%, 34.0%]) while two weeks delay cause a 21.6% (95% CI: [7.4%, 41.1%]) more intense incidence peak. The timing of the full lockdown also has a significant effect on the number of deaths. According to our estimates in Fig. 3b, just 1 week of delay implies a 7.7% (95% CI: [1.3%, 13.7%]) increase in mortality.

**Effective reproduction number**. In Fig. 3c we show the evolution of the effective reproduction number $R_t$ estimated using the method from ref. [40] on the simulated and the official reported incidence. In the simulated $R_t$ time series, we observe the two discontinuities after the implementation of government-issued NPIs. However, we note that the partial lockdown had the sole effect of slowing down the epidemic. Indeed, after March 16, the estimated $R_t$ is still >1. After the full lockdown, instead, $R_t$ was pushed below 1 making the containment possible. This is visible both in the simulated and the reported time series. This result underlines the importance of the full lockdown that, despite causing a relatively small effect on mobility, had a decisive role in bringing the outbreak under control. It is worth stressing this result. The full lockdown constituted a key tipping point for the evolution of the epidemic pushing the reproductive number below its critical threshold. A similar finding has been recently reported for the evolution of the pandemic in Germany[41]. Indeed, also in that context, only the subsequent compounding of interventions was able to bring the reproductive number below one thus curbing spreading of the virus.

# Discussion

The analysis presented here shows that the effects of NPIs issued by the government strongly correlate with a measure of human development, such as the HDI. In particular, comunas with higher HDI were able to reduce more significantly their mobility. This, in turn, is reflected in both data and modeling estimates by a lower burden of COVID-19 (i.e., cases, deaths) in the comunas characterized by a higher HDI. The combination of these results raises policy-making concerns. Indeed, while lockdowns are unquestionably effective in mitigating the epidemic activity, they may as well augment social and health inequalities, penalizing more vulnerable communities. Other studies have found that mobility restrictions unequally affected different regions of France[15], Italy[20], United States[19], Colombia, Mexico, and Indonesia[21] with a higher income being associated with a larger capacity to afford social distancing. Furthermore, observations in

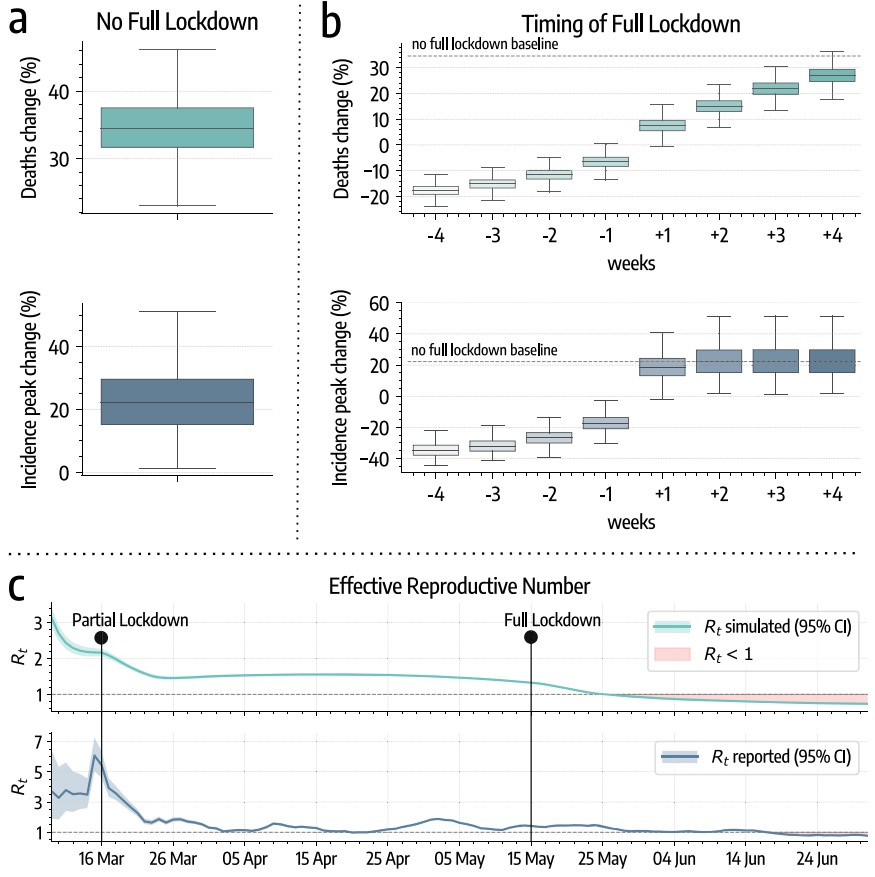

**Fig. 3 Impact of nonpharmaceutical interventions on COVID-19 spreading. a** Model estimates of percentage increases in deaths and incidence peak intensity without the implementation of the full lockdown (based on $n = 5000$ stochastic realizations). **b** Model estimates of percentage changes in deaths and in incidence peak intensity moving the date of the full lockdown of $-4/+4$ weeks (based on $n = 5000$ stochastic realizations). In both panels, the center of the boxes indicates the median, the bounds indicate the interquartile range (IQR) (i.e., the range between first quartile, Q1, and third quartile, Q3), and the whiskers indicate the minimum and the maximum defined respectively as $Q1 - 1.5IQR$ and $Q3 + 1.5IQR$. **c** Effective Reproductive Number $R_t$ estimated on simulated and officially reported cases. The two time series show a high positive Pearson correlation coefficient ($\rho = 0.78$, $p < 0.001$). The shaded red area indicates $R_t < 1$.

the United States[19,22,42,43], Singapore[44], and the UK[45,46] show that socioeconomic inequalities are linked to worst health outcomes during the current pandemic.

Our data-driven analysis also shows that the timeliness of NPIs is just one variable influencing the outcome of the mitigation effort. The case of Santiago is emblematic. NPIs were introduced early with respect to other countries. Only 2 days after the first 50 confirmed cases. For comparison, Denmark introduced measures after five, Austria after nine, Italy and Germany after 15 days of reaching that threshold[47]. These measures were followed by a considerable reduction in mobility and contacts, but cases soared anyway in the metropolitan area. According to our analysis, the first set of NPIs significantly slowed down transmissions but not enough to stop the epidemic. It was only after the introduction of the second additional lockdown that the Santiago outbreak was brought under control. This indicates that earlier implementation of more stringent NPIs may be beneficial in quickly mitigating the outbreak without extending for a long time policies that potentially might unequally affect communities. Similar results have been reported in the context of China[48], New Zealand[49], and the USA[8,50].

The present work comes with limitations. First, we focused only on the epidemic evolution within the Santiago Metropolitan Area and we overlooked both national and international importations after March 1, 2020. While it is reasonable to assume that

after this date the epidemic was largely sustained by the internal spreading (especially considering the various restrictions on international and national mobility), we acknowledge this as a possible limitation. Second, compared to other approaches[4,51,52], we considered a relatively simple disease dynamics. However, as shown in Section 2 of the Supplementary Information, our results are robust respect to the compartmental structure adopted. Third, our mobility measures also have limitations. Indeed, we considered the overall reduction in mobility and contacts. However, not all mobility translates equally in terms of transmission risk. Some types of locations such as for example restaurants and hotels have been linked to higher chances of infection than others[22]. We acknowledge that our mobile phone users' sample was not selected to be representative of the whole population. However, Telefónica Movistar data well represent the different socio-demographic groups of Santiago[53]. Last, mobile phone data streams are dependent on the distribution of antennas. Nonetheless, this issue is, at least partially, solved by the geographical level of aggregation we use here which is that of the comunas. In fact, for all 342 comunas of continental Chile, the Pearson correlation coefficient of census data and "home location" inferred from mobile phone data is 0.97 (more details in Section 5 of the Supplementary Information).

Overall, our study characterizes the unfolding of SARS-CoV-2 in one of the largest metropolitan areas in South America; a

region that so far has received far less attention than others. It quantifies the unequal effects across communities of behavioral changes introduced by governmental measures as well as individual (re)actions and provides evidence that even small delays in the implementation of NPIs can have a significant impact on the unfolding of the epidemic.

## Methods

**Measuring mobility and contacts.** In this work, we use phone data in the form of eXtended Detail Records (XDR). This stream records every interaction (e.g., packet request) between devices and antennas. An entry in our dataset can be formalized as a tuple $\langle d, t, a \rangle$ indicating a packet request to antenna $a$ made by device $d$ at time $t$. We approximate the position of $d$ with that of the antenna, which has a fixed latitude and longitude. We also assign a home antenna to each device by finding the most active antenna during night hours. Finally, we assign antennas to correspondent comunas according to their position. The dataset includes data for the period February 27, 2020 to June 1, 2020, for 1.4 million devices which correspond to the 22% of the total population in the area considered. To preserve the privacy of device owners, we analyze and display only anonymous and aggregated results. Furthermore, no other information about the users (i.e., gender, age etc.) was used or available.

As specified in the previous sections, by considering the governmental response and observing variations in the overall mobility, we identify three phases of nonpharmaceutical interventions and characterize them in terms of mobility and contacts reduction. Mobility is measured considering the fraction of devices traveling between comunas. Formally, for each day $t$ we build a mobility rate matrix $\boldsymbol{\sigma}(t) \in \mathbb{R}^{N \times N}$ whose element $\sigma_{ij}(t)$ is the fraction of devices living in comuna $i$ that visited $j$ on day $t$. It is important to mention how we do not have information about points of interest (POI). Hence, the data capture all types of mobility such as commuting for work, grocery runs, and/or recreational activities. We average these daily rates during the three phases to obtain three distinct matrices describing mobility (i) before any restrictions, (ii) during the partial lockdown, and (iii) during the full lockdown.

The data we use do not provide direct information about physical contacts between users. Having the privacy of the users in mind, and considering the various non-trivial assumptions one would need to make, we do not attempt to estimate/ infer such contacts. Instead, we focus on a metric that allows us to capture the variation before and after the various interventions. As we describe below, the epidemic model considers a homogeneous mixing approximation within each subpopulation (i.e., comuna) hence the only important variable is an estimate of contacts reduction rather than the actual contacts. To this end, contacts reduction is estimated by looking at the variation in the number of users co-located in the same antenna. Each antenna $a$ in comuna $j$ has a resident population $N_{a_j}$. On day $t$, the total number of visitors from the same comuna is $v_{a_j}(t)$. Assuming homogeneous mixing, the maximum number of contacts in antenna $a$ is $c_{a_j}(t) = (N_{a_j} + v_{a_j}(t)) \times (N_{a_j} + v_{a_j}(t) - 1)/2 \sim (N_{a_j} + v_{a_j}(t))^2/2$. Then, we assume the reduction of contact during the partial and full lockdown to be equal to:

$$r_{a_j}^{partial} = \frac{\underset{16/03 < t < 15/05}{avg} [c_{a_j}(t)]}{\underset{t<16/03}{avg} [c_{a_j}(t)]} \qquad r_{a_j}^{full} = \frac{\underset{t>15/05}{avg} [c_{a_j}(t)]}{\underset{t<16/03}{avg} [c_{a_j}(t)]} \qquad (1)$$

In other words, the reduction of contacts during the partial and full lockdown is considered as the variation of the maximum number of contacts before and after each intervention. Finally, we aggregate at the level of comunas taking the median of these quantities over all antennas located in the same comuna.

**Modeling the spread of SARS-CoV-2 in Santiago.** The model used in this work to simulate the spread of SARS-CoV-2 is largely inspired by the global epidemic and mobility model (GLEAM)[54–56]. In this section, we present the conceptual framework but a full mathematical description is provided in Section 4 of the Supplementary Information.

The comunas of Santiago are represented as distinct subpopulations forming a metapopulation network. Inside each one, we divide individuals into $K = 16$ 5-year age brackets respecting the demographic of different comunas[57] and we use the country-specific contact matrix from ref. [58] to define the rates at which different age groups mix with each other. Individuals are also divided into compartments according to their health status. We consider a SLIR (Susceptible, Latent, Infectious, Removed) compartmentalization setup. A similar approach has been used in several modeling studies in the context of COVID-19[14,59,60]. Interacting with Infectious, Susceptibles move to the Latent stage in which they are not infectious yet. Only after the latent period, Latent become Infectious. Last, Infectious transit to the Removed compartment at a rate inversely proportional to the infectious period. In Section 2 of the Supplementary Information we repeat the analyses for another compartmentalization that includes pre-symptomatic and asymptomatic transmission. The findings are not impacted by a different compartmental setup. Furthermore, the simpler SLIR scheme leads to a closer reproduction of the observed deaths.

Individuals can get the infection interacting with infected in their home and in other connected metapopulations. To model this aspect, we consider the mobility network previously introduced to describe an effective coupling (i.e., the strength of connection) between comunas. Technically, we use a time-scale separation technique and approximation[55,61] to define a "force of infection" $\lambda_j^k$ that expresses the infection rate for individuals in age group $k$ residing in comuna $j$:

$$\lambda_j^k = \frac{\lambda_{jj}^k}{1 + \sigma_j/\tau} + \sum_i \frac{\lambda_{ji}^k \sigma_{ji}/\tau}{1 + \sigma_j/\tau}, \qquad (2)$$

$\tau$ defines the timescale of mobility and $\sigma_j = \sum_{i \neq j} \sigma_{ji}$ is the total mobility rate of population $j$. The first term in Eq. (2) represents the contribution from active infections in comuna $j$, and the sum instead describes the effective contribution from cases in other connected comunas $i$. As mentioned above, the data we use capture all types of movements across comunas. Hence, the coupling between subpopulations accounts for mobility linked to work (i.e., commuting), grocery, and other activities. In our formulation, we assume that such movements take place over a timescale that is much smaller than the disease timescale and the temporal resolution of the epidemic data. This assumption is supported by the data. In fact, as shown in Supplementary Fig. 12, the average duration of trips outside the home comuna is 4.5 h and 85% of them take place within 8 h. Although users may travel outside of their comuna for more than 8 h, the probability of a trip to last more than 1 day is <3%. The time-scale separation approach allows us to integrate the faster dynamics—that is, the mobility—and consider their contribution to the spreading processes without simulating individual mobility patterns and thus considerably simplifying the computational costs of the model. It is important to notice how such time-separation approximation is exact only in the case $\tau^{-1} \to 0$. However, it holds as long as the faster timescale is shorter than the transition rates of the disease[61]. As mentioned in details below, for COVID-19 these are of the order of several days. We set the value of $\tau^{-1} = 1/3$ day to also account for commuting patterns. In Section 4 of the Supplementary Information we provide the full derivation and more details.

Government interventions are implemented by changing the mobility network on March 16 (partial lockdown) and again on May 15 (full lockdown) to reflect the corresponding variations in the coupling between comunas. Similarly, on these dates, we multiply the age contact matrix of each comuna by $r_j^{\text{partial}}$ and $r_j^{\text{full}}$, respectively. We do not explicitly account for school closure since its effect is already included in the changes inferred from mobile devices data.

The model is fully stochastic and transitions among compartments are simulated through chain binomial processes[62–64]. More in detail, on a given time step the number of individuals transiting from compartment $X_j^k$ to compartment $Y_j^k$ is extracted from a binomial distribution: $Pr^{\text{Bin}}(X_j^k, p_{X_j^k \to Y_j^k})$. In the main text, we present results for a latent period of 4 days and an infectious period of 2.5 days, which imply a generation time $T_G = 6.5$ days, in line with current estimates[65,66]. In Section 1 of the Supplementary Information we show a sensitivity analysis to these choices which do not substantially impact the findings. We simulate deaths considering the estimates of the infection fatality rate from ref. [34] and a delay $\Delta$ after the transition to the Removed compartment. This delay is included to account for the time span between isolation of acute cases (i.e., hospitalization), death, and official reporting, which can be more than 2 weeks[67].

Initial seeding is done using the projections of active cases on March 1, 2020, in the Metropolitan area of Santiago from ref. [14] and assigning infections to different comunas proportionally to the population distribution. The calibration is performed on weekly deaths using an Approximate Bayesian Computation (ABC) Rejection method[23,24]. At each step of the ABC algorithm, a set of parameters $\theta$ is sampled from a prior distribution and an instance of the model is generated using these parameters. Then, an output quantity $E'$ of the model is compared to the corresponding real quantity $E$ using a distance measure $s(E', E)$. If this distance is greater (smaller) than a predefined tolerance $\epsilon$, then the sampled set of parameters is discarded (retained). After a sufficient number of iterations, the distribution of accepted sets will approximate the posterior distribution of parameters $P(\theta, E)$ given the evidence $E$ from the data. In this work, we set a flat uniform prior on the parameters the basic reproduction number $R_0 \in [2, 4]$ (in steps of 0.02) and on the delay in deaths $\Delta \in [14, 21]$ days (in steps of 1 day). We perform calibration using the median absolute percentage error as a distance metric with a tolerance of 20% on weekly deaths (in the Supplementary Information we present a sensitivity analysis on this tolerance). We run 140,000 iterations which correspond to about 200 stochastic realizations for each possible parameter set. We use the official data issued by the Department of Statistics of the Chilean Minister of Health[68]. We consider both COVID-19 "confirmed" and "suspected" deaths to perform the calibration. In Supplementary Fig. 1 we support this decision showing that considering only confirmed COVID-19 deaths we still obtain a significant anomaly in mortality. Nonetheless, we also repeat the calibration only on "confirmed" deaths showing that the posterior distribution of $R_0$ is smaller but not significantly different. Model projections are produced sampling parameter sets directly from the posterior distribution and generating an ensemble of trajectories. In this work, we generate model estimates sampling 5000 sets on which we compute median and confidence intervals.

**Measuring socioeconomic differences**. We measure socioeconomic differences using the Human Development Index (HDI). The HDI is a coefficient that measures the level of achievement of key aspects of human development (including life expectancy, education, and per capita income) in the population considered[18]. It is released regularly for different countries by the United Nations in the Human Development Report[69]. In Chile, the last official calculation of the HDI at the level of comunas was done in 2000[70]. However, to rely on more updated estimates, we computed the HDI using census data for the period 2013–2015, following the guidelines provided in ref. [18]. These estimates have been previously used to study the socioeconomic determinants of mobility in Santiago[71]. The code used for the calculation of HDI can be found in ref. [72]. In Supplementary Fig. 4 we repeat the analyses for other socioeconomic indicators. In particular, we consider separately the components of the HDI, namely the Life Expectancy Index (LEI), the Education Index (EI), and the Income Index. The general pattern presented in the main text holds also for the other indicators considered.

**Reporting summary**. Further information on research design is available in the Nature Research Reporting Summary linked to this article.

## Data availability

The raw data analyzed in the study are not publicly available due to privacy reasons. All the aggregated mobility data needed to run the model are available at ref. [73]. Analysis of the anonymised mobile phone data was performed on mobile operator's systems without transferring it outside. Only aggregated mobility patterns across municipalities were provided to researchers outside Chile and only these have been used for the results presented here. The study was deemed exempt (IRB #20-10-05) by the Northeastern University Internal Review Board.

## Code availability

The main epidemic model described in the manuscript has been implemented with the programming language C++. The alternative models, detailed in Section 2 of the Supplementary Information, have been implemented in C++ or in Python. All codes are available at ref. [73], on GitHub https://github.com/ngozzi/covid19-santiago and on Zenodo https://zenodo.org/record/4607303.

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

## Acknowledgements

All authors thank the High Performance Computing facilities at Greenwich University. L.F. thanks Víctor Navarro and Loreto Bravo. N.G. acknowledges support from the Doctoral Training Alliance. M.T. acknowledges support from the Lagrange Project of the Institute for Scientific Interchange Foundation (ISI Foundation) funded by Fondazione Cassa di Risparmio di Torino (Fondazione CRT). LF acknowledges the funding and support of Telefónica R&D Chile and CISCO Chile. M.T. acknowledges support from EPIPOSE - "Epidemic intelligence to minimize COVID-19's public health, societal and economical impact" H2020-SC1-PHE-CORONAVIRUS-2020 call. M.C. and A.V. acknowledge support from Google Cloud and Google Cloud Research Credits program, the McGovern Foundation, and the Bill & Melinda Gates Foundation (award number INV006010).

## Author contributions

N.G., M.T., L.F., N.P., and A.V. designed the study. N.G. and N.P. analyzed the data. N.G. implemented and run the epidemic model. L.F. mined and provided the anonymised mobile phone data. M.C. provided the importation data to initialize the model. All authors interpreted the results, wrote and approved the manuscript.

## Competing interests

A.V. and M.C. report grants from Metabiota Inc., outside the submitted work; M.T. reports personal fees from GSK, outside the submitted work. All other authors declare no competing interests.
