## [Peer Review File · Nature Communications]

Reviewers' Comments:

Reviewer #1:

Remarks to the Author:

The authors built a meta-population model with a detailed age-structure to simulate the transmission of COVID-19 in Santiago de Chile. They investigated the effect of social inequalities on the non-pharmaceutical interventions against COVID-19. The method is well established, and the authors have applied similar models to many other infectious diseases (e.g., A/H1N1, Ebola, COVID-19). This manuscript needs a substantial improvement if the authors are aiming to publish it in Nature Communications. Otherwise, they may consider other more health policy focused journals.

Major comments:

(1) Readers will ask why Santiago de Chile is a very important location to study, and what new insights could be obtained by analyzing data of Santiago de Chile. This is too small scale a study to be globally applicable or even across the South American continent. There also lacks a comparison between Santiago de Chile and other places in Latin America.

(2) To simulate epidemic dynamics, the authors considered a SLIR compartment model for individuals within each subpopulation. They assumed that latent individuals will be infectious only after the incubation period. This assumption is acceptable for influenza or SARS. However, for COVID-19, it is known that a large proportion of infection (>40%) were due to pre-symptomatic transmissions. Therefore, the use of the SLIR model is expected to bias the estimated parameters. I strongly suggest the authors to use better disease models to simulate COVID-19, instead of using oversimplified compartment models.

(3) I'm not sure if it is suitable to use the regular commuting network with a fixed time scale of commuting (1/3 day) to model the human movement in South American countries such as Chile. In Latin America, a lot of people work in informal jobs without contracts. They may not commute on a regular basis. It will be helpful if the authors could use their mobile phone data to first build a suitable mobility model before building the epidemic metapopulation model.

(4) Regarding model fitting, the authors mentioned the use of the Approximate Bayesian Computation (ABC) approach. However, in the Methods section and supplementary materials, the authors did not provide any information to explain how to set up the ABC fitting method. As such, it is impossible to replicate their results, and readers will concern whether their method is correct.

(5) The use of complex metapopulation models may overfit the time series of death data. If a substantial proportion of individuals in Santiago de Chile has been infected, then the local infection may tend to follow simple well-mixing dynamics. The authors can fit the data using simpler models. It will be valuable to compare the performance of your complex model to simplified models. Model selection tools (e.g., out-of-sample cross-validation) will be needed. Then readers can better understand the contribution of your complex model.

(6) Prior settings often affect the posterior estimates. However, the authors did not clearly summarize their prior assumptions.

(7) In section 4.1, the authors stated "we characterize the three phases of the outbreak in terms of commuting and contacts reduction". Could you specify this point more clearly?

(8) In the last paragraph of page 8, "single subpopulations in a metapopulation network." What do you mean?

(9) Above section 4.3, how you design the "chain binomial processes"?

(10) Above section 4.3, "We simulate deaths considering the estimates of the Infection Fatality Rate from Ref. [19] and a delay after the transition to the Removed compartment". What is the delay distribution used? Do you have a comprehensive sensitivity analysis on the delay distribution?

(11) Readers may not be familiar with the Human Development Index (HDI). Could you give some discussion on why HDI but not other simpler socioeconomic index should be used to correlate with case counts? More sensitivity analysis using other socioeconomic indices would be needed.

Reviewer #2:

Remarks to the Author:

The manuscript titled "Estimating the effect of social inequalities on the mitigation of COVID-19 across communities in Santiago de Chile" is an insightful study on the impact of lockdown on the spread of COVID-19. Using relatively abundant mobility data, census data, and well-defined metrics, the authors quantified the reduction of commuting between comunas resulted by the lockdowns, the relations between commuting drops and socialdemographic factors, eventually estimated the R and simulated epidemics under different scenarios.

I recommend for publication, though there're two issues I would love to have the authors improve or discuss:

1. I had a hard time fully understanding the model structure and how the author derived the Eqn. 2 in the main text.

(1) The definitions for λ_j is inconsistent with those for other parameters, e.g., σ_j . The authors used "comunas j" in the main text, while "population j" in the supplementary information. I believe λ_j indicates the force of infection that individuals live in comunas j was infected in comunas i.

(2) Eqn. 3-6 in supplementary information: I think the authors used "(t)" to denote the time-dependent variables, and others without "(t)" as values/parameters; If so, Eqn. 6 is confusing: X_j , as a certain compartment in the stochastic SLIR model and the sum of two variables X_{jj} and X_{ji} , should be a time-dependent variable too. I understood, after a long time, that the authors first simulated the SLIR model, then regarded the S, L, I, R as values to derive the following equations. However, the notations are confusing and distracting without detailed interpretations, in both the main text and the supplementary information.

(3) Page 5 in supplementary information: by the definition of σ_j , isn't $\sum_j \sigma_j = 1$, since it also include σ_{jj} ?

(4) What's the reason for using the equilibrium value of X_{jj} and X_{ji} to derive the expression of λ_j and N_j ?

(5) If I don't get it wrong, the only parameters that the authors estimated using ABC and the metapopulation SLIR model is the transmission rate β , right?

2. For figure 2: it's a little uncommon to fit the model to the death data, instead of the reported cases. Intuitively, the number of infections is closely related to the contacts, while the number of deaths can be affected by factors like medical care level, etc. I wonder, can the authors compare the simulated trend of infections to the weekly reported confirmations? If not, can the authors discuss it?

Reviewer #3:

Remarks to the Author:

Thank you for the opportunity to read and review this interesting article. My comments include:

1. This article mentioned "real-time mobility" twice in the introduction but few information was given in the following sections. How "real-time mobility" was implemented using mobile-phone data? Is it through a real-time data stream APIs provided by "Telefonica Movistar"? If yes, what was the performance of conducting modelling from this real-time streaming data?

2. Figure 1B is quite interesting. I observed no change in commuting rates at the inner region (e.g., commute from Padre Hurtado to Padre Hurtado). Is it because no changes or current mobile-phone dataset cannot capture inner region changes? Also, what is the method or parameters to extract commuting travels from general travels?

3. Regarding the third limitation in the discussion section, the Point of Interest (POI) dataset could be very helpful to tackle this challenge.

4. The eXtended Detail Records (XDR) dataset seems like a classic mobile phone sightings dataset. If not, please verify. A major issue about this type of mobile-phone data is that the spatial resolution of data analysis is largely depends on the spatial distribution of antennas. Could authors provide general information such as what is the distribution of antennas? How often the a devices is recorded by a antennas in this dataset (e.g., 1 seconds? or 1 hour?)

5. Although the dataset is anonymous and no gender/age information was available, anonymous personal-level trajectories were still exposed to authors, which is forbidden in some countries by laws. If possible, the authors can provide additional ethical information e.g., what types of agreement was in place with "Telefonica Movistar", what was done to make sure individuals stay anonymous, what additional measures were taken to make sure each cell phone users are not identifiable.

6. In method, the "contact" was estimated by the number of users co-located in the same antenna, which is reasonable in many locations such as shopping mall, bus station and parks. However, this method is also problematic in residential areas. For example, 1K people stay at home all days during the lockdown. Also large number of users co-located in this antenna, they should have few social contact.

7. In the SLIR modelling, the choose of parameters is critical to simulation results. Although the parameters (e.g. 4 days incubation period, and 2.5 days infectious period) came from recent research, there are still debates. Authors should mentioned different chose of SLIR parameters many largely impact the simulation results in this research.

8. According to reference No.35, it seems that the Telefonica Movistar data can well represent the socio-demographic in Santiago. Does it introduce other bias? For example, is the spatial distributions of users proportional to the distribution of population?

Reviewer #1

We would like to thank the reviewer for the time spent reading and analyzing our paper. The constructive criticisms and suggestions raised have been important to improve the quality and clarity of the manuscript.

(1) Readers will ask why Santiago de Chile is a very important location to study, and what new insights could be obtained by analyzing data of Santiago de Chile. This is too small scale a study to be globally applicable or even across the South American continent. There also lacks a comparison between Santiago de Chile and other places in Latin America.

We thank the reviewer for this comment.

Even though it received less attention from the media with respect to other cities around the world, during the first wave of COVID-19, Santiago became one of the most affected urban areas globally. In fact, as of August 1, 2020, just in the Metropolitan Area of Santiago were reported more cases (256'628) than those reported in the whole Italy (247'832). Thus, even considering this point alone, we believe it is inherently important to characterize the spread of SARS-CoV-2 in one of the most affected cities of the world.

Santiago is characterized by marked social inequalities. Indeed, while being regarded as a high-income country, Chile is one of the most unequal. Unfortunately, this makes Santiago a natural case study to investigate the link between socio-economic disparities and the burden of the pandemic, which is one of the goals of the paper.

We would like to point out that research on similar “local” scales of study has been conducted and reported in the literature. Examples are the cases of Boston, Wanzhou, New York City and London. Indeed, while COVID-19 is a global issue, the measures put in place to mitigate or suppress its spread have been quite heterogeneous across countries as well as subregions within a country. Hence, we believe it is important to model and understand the effect of such non-pharmaceutical interventions in specific contexts as we do here.

We agree with the reviewer that these points need to be crystal clear. Hence, we added a more detailed discussion in the introduction.

(2) To simulate epidemic dynamics, the authors considered a SLIR compartment model for individuals within each subpopulation. They assumed that latent individuals will be infectious only after the incubation period. This assumption is acceptable for influenza or SARS. However, for COVID-19, it is known that a large proportion of infection (>40%) were due to pre-symptomatic transmissions. Therefore, the use of the SLIR model is expected to bias the estimated parameters. I strongly suggest the authors to use better disease models to simulate COVID-19, instead of using oversimplified compartment models.

The pre-symptomatic transmission is effectively accounted for by the choice of parameters regulating the generation time. In other words, the infection dynamics considered deals with the pre-symptomatic transmission since the infectious compartment includes both symptomatic and pre-symptomatic carriers (which we assume for simplicity to be equally infectious). This approach is found in a wide range of published work. Two examples are the following highly influential articles:

Zhou, Y., Xu, R., Hu, D., Yue, Y., Li, Q. and Xia, J., 2020. Effects of human mobility restrictions on the spread of COVID-19 in Shenzhen, China: a modelling study using mobile phone data. *The Lancet Digital Health*, 2(8), pp.e417-e424.

Tian, H., Liu, Y., Li, Y., Wu, C.H., Chen, B., Kraemer, M.U., Li, B., Cai, J., Xu, B., Yang, Q. and Wang, B., 2020. An investigation of transmission control measures during the first 50 days of the COVID-19 epidemic in China. *Science*, 368(6491), pp.638-642.

Including explicitly a pre-symptomatic compartment would be extremely important in approaches aimed at modeling contact tracing and isolation strategies, issues we are not addressing in the manuscript. Nevertheless, we also acknowledged this point as one of the limitations, the model we used is simple in comparison to others found in the literature.

Hence, following the reviewer’s suggestion, we extended our approach considering also a more complex disease dynamic. In particular, beside a SLIR model we now study a more refined compartmentalization.

Susceptible (S) individuals after interacting with infectious transit to the Latent compartment (L). After the latent period, L individuals enter the prodromal phase (P). P individuals then evolve either in the asymptomatic (A) or the symptomatic stage (I) (the length of time including L and P stages is the incubation period). Both I and A individuals after the infectious period enter the Recovered compartment (R). We compute deaths considering only on the Recovered resulting from the I compartment (i.e., symptomatic). The infectious compartments are P, A, I. We assume that P and I have lower infectiousness with respect to symptomatic I.

Similar approaches have been employed in other modeling studies. These are two examples:

Di Domenico, L., Pullano, G., Sabbatini, C.E. *et al.* Impact of lockdown on COVID-19 epidemic in Île-de-France and possible exit strategies. *BMC Med* 18, 240 (2020)

Hao, X., Cheng, S., Wu, D., Wu, T., Lin, X. and Wang, C., 2020. Reconstruction of the full transmission dynamics of COVID-19 in Wuhan. *Nature*, 584(7821), pp.420-424.

We set some of the key epidemiological parameters from the literature (which we cite in the main text and in the SI):

Latent period (time spent in E): 3.7 days
Prodromal stage (time spent in P): 1.5 days
Fraction of asymptomatic carriers: $r = 0.2, 0.4$
Ratio of transmission rate of I vs P, A infectious: $\alpha = 0.55$
Infectious period (time spent in A, I): 2.5 days

Ferretti, L., Wymant, C., Kendall, M., Zhao, L., Nurtay, A., Abeler-Dörner, L., Parker, M., Bonsall, D. and Fraser, C., 2020. Quantifying SARS-CoV-2 transmission suggests epidemic control with digital contact tracing. *Science*, 368(6491)

Lavezzo, E., Franchin, E., Ciavarella, C., Cuomo-Dannenburg, G., Barzon, L., Del Vecchio, C., Rossi, L., Manganelli, R., Loregian, A., Navarin, N. and Abate, D., 2020. Suppression of a SARS-CoV-2 outbreak in the Italian municipality of Vo'. *Nature*, 584(7821), pp.425-429.

Mizumoto, K., Kagaya, K., Zarebski, A. and Chowell, G., 2020. Estimating the asymptomatic proportion of coronavirus disease 2019 (COVID-19) cases on board the Diamond Princess cruise ship, Yokohama, Japan, 2020. *Eurosurveillance*, 25(10), p.2000180.

Li, R., Pei, S., Chen, B., Song, Y., Zhang, T., Yang, W. and Shaman, J., 2020. Substantial undocumented infection facilitates the rapid dissemination of novel coronavirus (SARS-CoV-2). *Science*, 368(6490), pp.489-493.

Backer, J.A., Klinkenberg, D. and Wallinga, J., 2020. Incubation period of 2019 novel coronavirus (2019-nCoV) infections among travellers from Wuhan, China, 20–28 January 2020. *Eurosurveillance*, 25(5), p.2000062.

Kissler, S.M., Tedijanto, C., Goldstein, E., Grad, Y.H. and Lipsitch, M., 2020. Projecting the transmission dynamics of SARS-CoV-2 through the postpandemic period. *Science*, 368(6493), pp.860-868.

The main findings obtained with the SLIR model hold also in the case of the more complex compartmentalization setup just described. The explicit addition of pre and asymptomatic transmission does not improve the fit with the data. While we present the more parsimonious model in the main text In the SI we now include the results of the simulations considering the more complex compartmentalization of the disease stages. We are thankful to the reviewer for proposing this important check/modification which undoubtedly adds value to the paper.

(3) I'm not sure if it is suitable to use the regular commuting network with a fixed time scale of commuting (1/3 day) to model the human movement in South American countries such as Chile. In Latin America, a lot of people work in informal jobs without contracts. They may not commute on a regular basis. It will be helpful if the authors could use their mobile phone data to first build a suitable mobility model before building the epidemic metapopulation model.

It is important to stress how the mobility data we use takes into account *all* movements by the users in the area of Santiago. Commuting (intended as work or school related mobility) will definitely be there but we also capture other types of mobility.

Our modeling approach is based on the assumption that movements relevant for the epidemic spread take place over a time-scale that is shorter than the time-scale marking the progression of the simulations, the disease and the temporal resolution of the epidemic data.

In the Supplementary Information we have added a plot that confirms this: the average duration of trips outside the home comuna is 4.5 hours. Furthermore, 85% of such trips take place within 8 hours. Although users may travel outside of their comuna for more than 8 hours, the probability of a trip to last more than 1 day is less than 3%.

Hence, the effects of human mobility on the epidemic can be captured through an effective force of infection (see Keeling & Rohani. Estimating spatial coupling in epidemiological systems: a mechanistic approach. Ecology Letters 2002).

The expression of the force of infection assumes that the dynamics of the coupling between comunas (i.e., movements) is faster, and it can be considered at equilibrium with respect to the dynamics that describe the spread of the disease (i.e., transition rates).

Mobility data that have been used to feed similar epidemic models is indeed often based on commuting which takes place within $\frac{1}{3}$ day. However, this can vary depending on the type of data available. Other mobility types taking place within the same time-scale (or even at faster time-scales) can be factored in without any change to the formulation of the model.

We used the word “commuting” too liberally in the initial submission. In the revised version of the manuscript, we make this important point clear and speak about mobility between comunas rather than “commuting” which better characterize our data. Furthermore, we now explain in more details the assumptions, approximations and the limits of validity of the model in the Methods.

(4) Regarding model fitting, the authors mentioned the use of the Approximate Bayesian Computation (ABC) approach. However, in the Methods section and supplementary materials, the authors did not provide any information to explain how to set up the ABC fitting method. As such, it is impossible to replicate their results, and readers will concern whether their method is correct.

We thank the reviewer for pointing this out.

Regrettably we realized that in the formatting process we missed to paste this section. We use the ABC rejection algorithm to find the posterior distribution of the two free parameters: the basic reproductive number and the delay in deaths after the transition to the Removed compartment. We set on both parameters a flat uniform prior. More in detail, we explore values of R_0 between 2 and 4, and values of Delta between 14 and 21 days. As a distance metric we use the median absolute percentage error with a tolerance of 20%. We run 140'000 iterations which correspond to about 200 stochastic realizations for each possible parameter set.

We amended the mistake. In the Materials and Methods section of the revised version, we included the details about ABC Calibration.

(5) The use of complex metapopulation models may overfit the time series of death data. If a substantial proportion of individuals in Santiago de Chile has been infected, then the local infection may tend to follow simple well-mixing dynamics. The authors can fit the data using simpler models. It will be valuable to compare the performance of your complex model to simplified models. Model selection tools (e.g., out-of-sample cross-validation) will be needed. Then readers can better understand the contribution of your complex model.

We respectfully disagree with the idea that the model is overfitting the time series of death data. In fact, the model has only two free parameters: the basic reproductive number and the delay in deaths after the transition to the Removed compartment. Except for the

epidemiological parameters borrowed from the literature, all other elements such as coupling between comunas and contact reductions within them are observed from data. Thus, the model simulates the spreading of the disease in the system given the set of parameters, empirical mobility and contacts rates.

Nevertheless, following the reviewer's suggestion, we also run a simplified model. In particular, we model each comuna as a single population disregarding commuting and we use the data-driven contacts reduction parameters. We include the results in the SI. We obtain a much worse fit of the data, indicating that the mobility network is actually important in modeling the spread of COVID-19 in the area considered. These results, we believe, clearly show the importance of accounting for mobility patterns to capture the unfolding of the outbreak.

Following the reviewer's comment further, we thought to implement an even simpler model considering the whole area, hence the 37 comunas, as a single population. This approach has been used quite often in the literature to model the spread of SARS-CoV-2 in cities, regions, and countries. However, the model would be agnostic about the differences in disease burden between comunas. Understanding such heterogeneities/differences, which are also clear in the epidemiological data, is a key aspect of our study hence we opted for skipping this approach.

Finally, we would like to stress that the goal of the paper is not producing forecasts or projections about the spreading of SARS-CoV-2 in the area. Our aim is focused on developing an understanding of the outbreak capturing, retrospectively, the first wave and identifying the effects of non-pharmaceutical interventions and social inequalities. Hence, we believe that out-of-sample cross validation approaches, though key when comparing/testing predictive frameworks, are beyond the scope of our research.

(6) Prior settings often affect the posterior estimates. However, the authors did not clearly summarize their prior assumptions.

We completely agree with the reviewer, as stated in point (4) we added this information. We set on both parameters a flat uniform prior. More in detail, we explore values of R_0 between 2 and 4, and values of Delta between 14 and 21 days.

(7) In section 4.1, the authors stated "we characterize the three phases of the outbreak in terms of commuting and contacts reduction". Could you specify this point more clearly?

By considering the governmental response and observing variations in the overall mobility we identified three phases of non-pharmaceutical interventions. Before 16/03 we have the, business as usual, baseline. Between 16/03 and 15/05 the first set of NPIs interventions was put in place. After 15/05 the metropolitan area was put in full lockdown. These three phases translate in the model in three different regimes of mobility among comunas, and contacts reduction within them. In other words, we have three mobility matrices and contact reduction rates for each phase. The details are described in section 2, material and methods section and in the SI.

(8) In the last paragraph of page 8, "single subpopulations in a metapopulation network." What do you mean?

In that section we describe the metapopulation network which is formed by subpopulations (i.e., comunas) connected by means of mobility. The word "single" is probably confusing but was there to highlight how each comuna is considered a subpopulation without any other

stratification except for the age-structure. In the revised version of the manuscript, we clarified the sentence

(9) Above section 4.3, how you design the “chain binomial processes”?

We adopted the classic stochastic approach used in compartmental models. Given the set of parameters which are either static (i.e., recovery rate) or dynamic (i.e., force of infection) the transitions between compartments are modelled with chains of binomial extractions modulated by the number of individuals in each compartment and the transition rates. We added further clarification to this statement together with some general references.

(10) Above section 4.3, “We simulate deaths considering the estimates of the Infection Fatality Rate from Ref. [19] and a delay after the transition to the Removed compartment”. What is the delay distribution used? Do you have a comprehensive sensitivity analysis on the delay distribution?

Unfortunately, this point was cut from the manuscript due an editing mistake. We used a flat uniform prior between 14 and 21 days for the delay in deaths and we fit it through ABC calibration.

We added this information.

(11) Readers may not be familiar with the Human Development Index (HDI). Could you give some discussion on why HDI but not other simpler socioeconomic index should be used to correlate with case counts? More sensitivity analysis using other socioeconomic indices would be needed.

In the revised version of the manuscript, we extended the paragraph “Measuring Socioeconomic Differences” in Materials and Methods section adding further details on the HDI and on its usage. Furthermore, the Supplementary Information includes sensitivity analysis regarding the correlation of mobility changes and other socio-demographic indicators, such as the Life Expectancy Index, the Education Index, and the Income Index. Our findings are consistent also using these different indices.

Reviewer #2

We would like to thank the reviewer for their detailed reading of the manuscript. The comments and suggestions have helped improve the manuscript.

The manuscript titled "Estimating the effect of social inequalities on the mitigation of COVID-19 across communities in Santiago de Chile" is an insightful study on the impact of lockdown on the spread of COVID-19. Using relatively abundant mobility data, census data, and well-defined metrics, the authors quantified the reduction of commuting between comunas resulted by the lockdowns, the relations between commuting drops and socialdemographic factors, eventually estimated the R , and simulated epidemics under different scenarios.

I recommend for publication, though there're two issues I would love to have the authors improve or discuss:

1. I had a hard time fully understanding the model structure and how the author derived the Eqn. 2 in the main text.

(1) The definitions for λ_{ji} is inconsistent with those for other parameters, e.g., σ_{ji} . The authors used "comunas j" in the main text, while "population j" in the supplementary information. I believe λ_{ji} indicates the force of infection that individuals live in comunas j was infected in comunas i.

We thank the reviewer again for the time spent understanding our work. We apologize for the source of confusion.

The interpretation of the force of infection is correct. We (implicitly) used the terms "population" and "comuna" as synonyms. In the revised version we used only the term "comuna" to avoid confusion

(2) Eqn. 3-6 in supplementary information: I think the authors used "(t)" to denote the time-dependent variables, and others without "(t)" as values/parameters; If so, Eqn. 6 is confusing: X_j , as a certain compartment in the stochastic SLIR model and the sum of two variables X_{jj} and X_{ji} , should be a time-dependent variable too. I understood, after a long time, that the authors first simulated the SLIR model, then regarded the S, L, I, R as values to derive the following equations. However, the notations are confusing and distracting without detailed interpretations, in both the main text and the supplementary information.

We thank the reviewer for pointing out the confusion on this point.

The time-scale marking the evolution of the simulations, the progression of the disease, and the temporal resolution of the epidemic data is a day. However, the mobility patterns we observe from data take place at a faster pace. For example, commuting for work is typically considered as $\frac{1}{3}$ of a day. Movements linked to grocery runs and other activities are even faster. In the Supplementary Information we have added a plot to support this intuition. In particular, we show the duration of trips outside home comunas. Interestingly, the average is 4.5 hours and 85% of such trips take place within 8 hours. Although users may travel outside of their comuna for more than 8 hours, the probability of a trip to last more than one day is less than 3%.

In our model we adopt a time-scale separation technique and approximation to integrate the faster dynamics (i.e., mobility) estimating their effective contributions to the slower processes (i.e., progression of the disease).

The "(t)" causing confusion describes times **within** a day. Quantities such as X_j are obtained integrating the effects of mobility (i.e., faster dynamics) over such times and thus are considered at equilibrium within a day. In doing so, we estimate the **effective** contributions to the force of infection from visitors and locals without having to simulate their actual movements within the day.

This approximation was originally introduced by Keeling and Rohani (Keeling M J, Rohani P, 2002, Estimating spatial coupling in epidemiological systems: a mechanistic approach. Ecology Letters 5: 20–29.), and allows to consider each subpopulation j as having an effective number of individuals X_{ji} in contact with the individuals of the connected subpopulation i.

The mobility time scale is separated from the other time scales (i.e., disease dynamics). The approximation is exact only in the case of infinitely fast dynamics. However, it holds as long as the faster time-scale is much smaller than the typical transition rates of the disease dynamics. For COVID-19, as well as for other diseases, these are on the order of days.

We realized this point was far from clear in the first version of the manuscript. Hence, in the Materials and Methods we now provide a more detailed discussion about the force of infection and the ideas behind the derivation such as the time-scale separation. Furthermore, in the SI, we added a more streamlined and clear derivation of all the quantities.

(3) Page 5 in supplementary information: by the definition of σ_{ji} , isn't $\sum_j \sigma_{ji} = 1$, since it also include σ_{jj} ?

We thank the reviewer for pointing out the confusion on this point. Actually, $\sum_j \sigma_{ji}$ is in general different from 1.

Indeed, intra-comuna mobility is not considered in the calculation of the force of infection. Each node of the metapopulation network is a comuna, therefore we are interested only in movements between different comunas, while we consider within a single comuna a mixing dynamics modulated by the contact matrices.

More precisely we defined σ_{ji} as “the fraction of devices living in comuna j that visited i on a day t ”. Hence, in general, this fraction is smaller than the total population of each comuna

We clarified the equation and fixed the notation explicitly excluding j from the summation in the new version of the SI.

*(4) What's the reason for using the equilibrium value of X_{jj} and X_{ji} to derive the expression of λ_j and N^*_j ?*

As mentioned in more details above, the basic idea behind the computation is to derive an expression for the force of infection in each subpopulation accounting for the effective contribution of infectious individuals from other comunas. To this end, we assume that mobility takes place at a faster time-scale with respect to the progression of simulations and disease (day). Hence, we consider the equilibrium values obtaining an effective expression which allows us to avoid considering “fractional” time-steps (to account for transients) within each day. We have added an explanation about this point on the Material and Methods sections and in section 3 of the SI

(5) If I don't get it wrong, the only parameters that the authors estimated using ABC and the metapopulation SLIR model is the transmission rate β , right?

We fit both the transmission rate β and the delay in reported deaths Δ . In the Materials and Methods section, we added details on the fitting procedure to specify all the details.

2. For figure 2: it's a little uncommon to fit the model to the death data, instead of the reported cases. Intuitively, the number of infections is closely related to the contacts, while the number of deaths can be affected by factors like medical care level, etc. I wonder, can the authors compare the simulated trend of infections to the weekly reported confirmations? If not, can the authors discuss it?

While some articles consider cases rather than deaths, the most recent trends in the literature lean more towards the use of confirmed deaths and/or hospitalizations. In fact, while there are biases in any indicator, the number of confirmed cases is arguably one of the most affected by varying reporting rates. Testing capabilities and testing strategies that

target only severe symptomatic individuals induce high levels of underreporting which are also time dependent.

Though not perfect, deaths/hospitalizations are less prone to underreporting than infections. We added a sentence to make this clear in the SI.

In Fig. 2C we show that, while the simulated number of infections well correlates with the official number reported by the Ministry of Health, we also note that the simulated number is much higher than the official one. This is not uncommon in the context of COVID-19. Indeed, seroprevalence studies conducted, for example, in the United States, Spain, Italy, Brazil, and Iran showed that the actual number of COVID-19 infections is several times (factors vary from 4 to 20) those reported by the official surveillance. We discuss this aspect also in Section 2.2.

Reviewer #3

We would like to thank the reviewer for the careful read and analysis of our work. The comments and suggestions have been very useful to clarify and improve the manuscript.

Thank you for the opportunity to read and review this interesting article. My comments include:

1. This article mentioned “real-time mobility” twice in the introduction but few information was given in the following sections. How “real-time mobility” was implemented using mobile-phone data? Is it through a real-time data stream APIs provided by “Telefonica Movistar”? If yes, what was the performance of conducting modelling from this real-time streaming data?

We thank the reviewer for noticing this. We have deleted references to “real-time mobility” in the manuscript since it’s not exactly real-time. In any case, so as to satisfy curiosity: we have mobile phone data automatically copied to a shared repository, which is then loaded to a cloud instance of a column-store database with secret keys managed by Telefonica. The stream deposits data every day, with a two-day lag. For example, on Tuesday, December 15 there is a new batch of data up until, and including, Sunday, December 13; on Wednesday, December 16, there is a new batch up until December 14, and so on.

2. Figure 1B is quite interesting. I observed no change in commuting rates at the inner region (e.g., commute from Padre Hurtado to Padre Hurtado). Is it because no changes or current mobile-phone dataset cannot capture inner region changes? Also, what is the method or parameters to extract commuting travels from general travels?

Intra-comuna mobility is not considered. Each node of the metapopulation network is a comuna, therefore we are interested only in mobility between different populations, while we consider within a single population a homogeneous mixing dynamic. As described in section 4.1 we use travels within the same comuna to estimate contact changes. Given the structure of the data, we are not able to distinguish commuting travels from general travels.

3. Regarding the third limitation in the discussion section, the Point of Interest (POI) dataset could be very helpful to tackle this challenge.

We completely agree with the reviewer, but this is, unfortunately, not available. Hopefully, we will be able to add this dimension in future work.

4. The eXtended Detail Records (XDR) dataset seems like a classic mobile phone sightings dataset. If not, please verify. A major issue about this type of mobile-phone data is that the spatial resolution of data analysis largely depends on the spatial distribution of antennas. Could authors provide general information such as what is the distribution of antennas? How often are devices recorded by antennas in this dataset (e.g., 1 second? or 1 hour?)

XDRs are **one** of the mobile phone streams that telcos have access to. It is one order or magnitude more temporally fine grained than Call Detail Records, the real “classic” mobile phone stream, and one order or magnitude less fine-grained than the control plane stream, which records all the network events associated with a device.

However, as the reviewer points out, all these streams are dependent on the distribution of antennas. Antennas are distributed by “demand” (more antennas where there’s more demand for signal, more phones at certain times of the day), and to a lesser extent by coverage (not leaving certain areas without mobile phone signal, like in rural areas).

Devices are recorded once every 15 minutes or 30 minutes (depending on the Base Transceiver Station technology), or after ~30MB have been downloaded. This effectively means that there is no overestimation of trips where antennas are denser.

Our mobility data suffers from the same limitations of the rest of the literature deriving mobility from mobile phone data (except maybe GPS, which is not done by Telcos but by apps). However, some of the issues mentioned by the reviewer are, at least partially, solved by the geographical level of aggregation we use here which is that of comunas.

We added a point about this in the limitations.

5. Although the dataset is anonymous and no gender/age information was available, anonymous personal-level trajectories were still exposed to authors, which is forbidden in some countries by laws. If possible, the authors can provide additional ethical information e.g., what types of agreement was in place with “Telefonica Movistar”, what was done to make sure individuals stay anonymous, what additional measures were taken to make sure each cell phone users are not identifiable.

Privacy and confidentiality are always of utmost importance for us as researchers and Telefónica. The dataset the telco shares with us is a tuple with the anonymized phone number (hashed), the latitude and longitude of the *tower* where the transaction took place (not the azimuth of the antenna mounted on that tower, for example, which makes trips even more underdetermined) and the timestamp. Also, only one of the authors (affiliated with Telefonica R&D) had access to the anonymized dataset. The access and mining of the data follow strictly the Chilean laws and the privacy preserving standards. Only aggregated mobility patterns across municipalities were provided to researchers outside Telefonica and only these have been used for the results presented. Together with the fact that there isn’t any demographic or other individual information, the study was deemed exempt (IRB #20-10-05) by the Northeastern University Internal Review Board.

6. In method, the “contact” was estimated by the number of users co-located in the same antenna, which is reasonable in many locations such as shopping mall, bus station and parks. However, this method is also problematic in residential areas. For example, 1K people stay at home all days during the lockdown. Also large number of users co-located in this antenna, they should have few social contact.

We completely agree with the reviewer. However, we do not have locations of POIs (such as shopping malls) in our dataset. Nevertheless, we point out that our definition of contacts reduction is i) a ratio (therefore it is simply a relative reduction in contacts), and ii) it is made of a contribution from the local population and possible visitors (therefore the possible problem pointed out by the reviewer should be accounted by our definition).

7. In the SLIR modelling, the choose of parameters is critical to simulation results. Although the parameters (e.g. 4 days incubation period, and 2.5 days infectious period) came from recent research, there are still debates. Authors should mentioned different chose of SLIR parameters many largely impact the simulation results in this research.

In the Supplementary Information, we include sensitivity analysis on the epidemiological parameters. We show that changing the epidemiological parameters do not substantially impact the findings. We added a mention to this sensitivity analysis in the main text.

8. According to reference No.35, it seems that the Telefonica Movistar data can well represent the socio-demographic in Santiago. Does it introduce other bias? For example, is the spatial distributions of users proportional to the distribution of population?

For all 342 comunas of continental Chile, the Pearson correlation coefficient of census data and “home location” as described in the main manuscript, is R^2 value: 0.96. We added this information in the discussion and a detailed plot in the SI.

Furthermore, mobile phone data penetration is very high in Chile: 136 devices over 100 people according to the Subsecretary of Transport and Telecommunications, and smartphones are universally available, together with free “bags of data” and most applications like whatsapp, instagram, facebook, twitter (though not Spotify or Netflix) are free. There are, surely, other biases as we mention in our limitations, but none of them are obvious enough and are not unlike the ones found in other similar studies.

Reviewers' Comments:

Reviewer #1:

None

Reviewer #2:

Remarks to the Author:

I would love to thanks the authors for carefully reading and addressing my comments and revising the manuscript. All of my comments are well addressed and I recommend for publication.

Reviewer #3:

Remarks to the Author:

My concerns regarding the research have been addressed. However, the 5th comment from reviewer #1 should be carefully looked into.

Reviewer #2

We would like to thank the reviewer for their detailed reading of the revised manuscript.

1) I would love to thank the authors for carefully reading and addressing my comments and revising the manuscript. All of my comments are well addressed and I recommend for publication.

We are glad our revisions addressed all points raised by the referee. We would like to thank the referee one more time for all comments, suggestions, and constructive criticisms as they really help improve the manuscript.

Reviewer #3

We would like to thank the reviewer for the time spent reading our revisions.

1) My concerns regarding the research have been addressed. However, the 5th comment from reviewer #1 should be carefully looked into.

We are happy that our revisions addressed the comments raised by the referee in the first round. They were all very helpful and allowed us to improve our work.

The 5th comment from reviewer #1 was:

(5) The use of complex metapopulation models may overfit the time series of death data. If a substantial proportion of individuals in Santiago de Chile has been infected, then the local infection may tend to follow simple well-mixing dynamics. The authors can fit the data using simpler models. It will be valuable to compare the performance of your complex model to simplified models. Model selection tools (e.g., out-of-sample cross-validation) will be needed. Then readers can better understand the contribution of your complex model.

To address these points, in the first revision of the paper, we have run a simplified model. In particular, we considered each comuna as a single population disregarding commuting and we used the data-driven contacts reduction parameters. We found a much worse fit of the data, indicating that the mobility network is actually important in modeling the spread of COVID-19 in the area considered.

While the referee does not suggest any specific points we should be looking into more carefully, the comment suggests the need for further analysis.

To this end, we have implemented two additional simpler models. Following approaches that have been used quite often in the literature to model the spread of SARS-CoV-2 in cities, regions, and countries both models consider the whole metropolitan area as a single, age-structured, population. The first, adopts the mobile phone data to estimate the variations to the contact matrices as a function of the non-pharmaceutical interventions. In the second instead, such variations are estimated using the Google Mobility Reports and the Oxford COVID-19 Government Response Tracker. The first model performs much better than the

second. However, it is important to mention how its performance is inferior to the main model proposed in the article. Furthermore, this simpler model is agnostic to the differences in disease burden between comunas. Understanding such heterogeneities/differences, which are also clear in the epidemiological data, is a key aspect of our study.

Overall, the comparison of our main model with these three simpler approaches confirms how capturing the spatio-temporal spreading of the virus within and across comunas is crucial to obtain a deeper and more precise description of the pandemic in the area

As already mentioned in the first round of revision, the goal of the paper is not producing forecasts about the spreading of SARS-CoV-2 in the area. Our aim is focused on developing an understanding of the outbreak by capturing, retrospectively, the dynamics of the first wave and by identifying the effects of non-pharmaceutical interventions and social inequalities. Hence, we believe that out-of-sample cross validation approaches, though key when comparing/testing predictive frameworks, are beyond the scope of our research.